# Processes Controlling the Contractile Ring during Cytokinesis in Fission Yeast, Including the Role of ESCRT Proteins

**DOI:** 10.3390/jof10020154

**Published:** 2024-02-15

**Authors:** Imane M. Rezig, Wandiahyel G. Yaduma, Christopher J. McInerny

**Affiliations:** 1School of Molecular Biosciences, College of Medical, Veterinary and Life Sciences, University of Glasgow, Davidson Building, Glasgow G12 8QQ, UK; wandiahyel.yaduma@glasgow.ac.uk; 2Department of Chemistry, School of Sciences, Adamawa State College of Education, Hong 640001, Adamawa State, Nigeria

**Keywords:** cell cycle, cytokinesis, contractile ring, fission yeast, ESCRT

## Abstract

Cytokinesis, as the last stage of the cell division cycle, is a tightly controlled process amongst all eukaryotes, with defective division leading to severe cellular consequences and implicated in serious human diseases and conditions such as cancer. Both mammalian cells and the fission yeast *Schizosaccharomyces pombe* use binary fission to divide into two equally sized daughter cells. Similar to mammalian cells, in *S. pombe*, cytokinetic division is driven by the assembly of an actomyosin contractile ring (ACR) at the cell equator between the two cell tips. The ACR is composed of a complex network of membrane scaffold proteins, actin filaments, myosin motors and other cytokinesis regulators. The contraction of the ACR leads to the formation of a cleavage furrow which is severed by the endosomal sorting complex required for transport (ESCRT) proteins, leading to the final cell separation during the last stage of cytokinesis, the abscission. This review describes recent findings defining the two phases of cytokinesis in *S. pombe*: ACR assembly and constriction, and their coordination with septation. In summary, we provide an overview of the current understanding of the mechanisms regulating ACR-mediated cytokinesis in *S. pombe* and emphasize a potential role of ESCRT proteins in this process.

## 1. The Use of Fission Yeast to Study Eukaryotic Cytokinesis

Both mammalian cells and the fission yeast *Schizosaccharomyces pombe* use binary fission to divide medially. Fission yeast cells are encased in a cell wall structure, giving them their rod shape following growth by tip extension, and divide equatorially. Therefore, the species is considered an excellent model organism for studying eukaryotic cytokinesis, during which similar cellular processes occur. 

Research using *S. pombe* has allowed for the identification of many important conserved cell cycle regulators. As the mechanisms of the assembly and constriction of the actin–myosin contractile ring (ACR) in *S. pombe* are very similar to those seen in mammalian cells, our current understanding of eukaryotic cytokinesis has benefited significantly from studies in fission yeast [1].

During the 1970s, the work by Hartwell and colleagues with the budding yeast *Saccharomyces cerevisiae* led to the discovery of a large number cell division cycle (CDC) mutants, and for the first time, the eukaryotic genes required for cell division were characterized [2,3]. The study of the cell division cycle continued in the distally related fission yeast *S. pombe* by Nurse and colleagues, with the discovery of equivalent *cdc* gene mutants [4,5]. 

Later, in the 1990s, two landmark reviews discussed aspects of the *S. pombe* cell cycle including the timing of events leading to cytokinesis, cell division and mechanisms for determining the medial or equatorial division plane. At that time, the *cdc16* and *cdc2* genes were thought to act as a molecular switch regulating *S. pombe* mitosis and cytokinesis [6], and it was proposed that the division plane was determined by the position of the nucleus [7]. Subsequent research offered a deeper understanding of the *S. pombe* cell cycle regulation, including aspects of cytokinesis, reviewed in Nurse et al. [8], and cell polarity, reviewed in Chang et al. [9].

The establishment of cytokinesis is mediated by a cytokinetic ACR that leads to the final separation of the two daughter cells. In this short review, we describe *S. pombe* cytokinesis starting from the medial positioning of the division plane and the assembly of the ACR (Section 2), to the forces that generate tension and lead to the constriction of the ACR and the role of ESCRT proteins in cytokinesis, to the final separation of the two daughter cells following septation (Section 3 and Section 4).

## 2. Actin–Myosin Contractile Ring (ACR) Assembly in Fission Yeast

### 2.1. Positioning of the Cell Division Plane

In *S. pombe*, cellular growth occurs throughout a longer interphase period, with this ceasing during the shorter mitosis and cytokinesis periods after a certain cell length is achieved. During the cell cycle, the “middle” and “end or tip” locations are specified by two spatial axes. The “middle” location is defined by the nucleus, which is positioned at the cell center by a microtubule-pushing mechanism, where a force is produced by the cytoplasmic microtubule bundles and acts on the nucleus [10,11]. Furthermore, this force is able to efficiently re-center the nuclei of cells exposed to nuclear displacement [12,13]. The dynamic interplay between the nucleus and the microtubule cytoskeleton is illustrated and reviewed by Gallardo et al. [14]. The “end or tip” location is defined by a formin-mediated actin assembly mechanism at cell tips [15], and polarity factors including the DYRK kinase Pom1p gradient at cell poles [16]. Pom1p gradients are tightly controlled at the “end or tip”, with the dephosphorylation of Pom1p enhancing a lipid-binding activity, whereas autophosphorylation promotes Pom1p’s detachment from the plasma membrane [16]. Pom1p has an established role in regulating the timing of mitotic entry, as it phosphorylates the membrane-binding C-terminal region of the ACR scaffold protein, Cdr2p, at the cell “middle” through preventing its plasma membrane binding and the formation of nodes [17]. Research has revealed a strong correlation between the cell size at division and Pom1p medial levels; however, such a correlation between the cytosolic or cell tip levels of Pom1p and the cell length is inconsistent, indicating that Pom1p may interact with Cdr2p in the cytosol or at the cell tips [18]. For example, the Pom1p gradient model [16] is opposed by the findings of Pan et al. [19], in which they proposed a novel cell size control mechanism in which cells use Cdr2p to monitor their size. This model therefore implicates a relationship between both Pom1p and Cdr2p with cell size.

Much evidence shows that the Anillin-like protein Mid1p localizes to the “middle” location and initiates ACR assembly [20,21,22,23]. Mid1p has two membrane binding domains, the pleckstrin homology domain (PH) and the cryptic domain (C2) [24]. However, it only binds the plasma membrane after it is activated and released from the nucleus [20,25]. The roles of Mid1p in positioning the ACR are now well understood in *S. pombe* and are reviewed in Rezig et al. [26], with the mechanism of the medial positioning of the ACR schematically described in Figure 1.

In contrast, the comprehension of the organization of proteins that assemble the complex cytokinetic machinery during cytokinesis is still relatively rudimentary. In *S. pombe*, cytokinesis proteins are recruited to the cell center, pre-determining the future division plane; these are organized as cortical spots, named “nodes” [27,28]. The next section will discuss the nature of these nodes including their constituent proteins and spatiotemporal organization. 

### 2.2. Molecular Organization of Nodes within the ACR

The current model for ACR assembly includes the formation of two types of interphase nodes: type 1 “stationary” nodes containing Mid1p, Cdr1p and Cdr2p; and type 2 “anchoring” nodes containing Blt1p, Klp8p, Gef2p and Nod1p [27,29]. Type 2 “anchoring” nodes diffuse into the cell cortex, and at mitotic onset, they are captured by type 1 “stationary” nodes to form cytokinesis nodes. Next, Mid1p recruits the myosin-II, Myo2p, Cdc15p, Rng2p and Cdc12p [30]. The cytokinesis nodes then merge into a ring-like structure, named the actin–myosin contractile ring (ACR) and, as its name implies, it is composed of actin filaments and myosin-II motors in addition to various classes of cytokinesis proteins [31]. 

Live cell imaging, high-speed fluorescence photo-activation localization microscopy (FPALM), and fluorescence resonance energy transfer (FRET) have been shown to be excellent methods to dissect ACR nodes. Recent findings have revealed that nodes are discrete units with stoichiometric ratios and a specific distribution of constituent proteins [28,30,31,32]. Furthermore, the localization of the ACR constituents is thought to be arranged in several layers relative to the plasma membrane, starting with the plasma-membrane-binding proteins and the tail of myosin-II, to the intermediate cytokinesis proteins, and farthest from the plasma membrane lies the myosin motor domains, F-actin and its cross-linkers [32]. Advances in laser scanning microscopy, such as Airyscanning using a very low laser power to acquire high-quality images, have increased the resolution and signal-to-noise ratio and enabled the detection and measurement of even faint individual cytokinesis nodes [33]. The coalescence of nodes leads to ACR assembly through the search, capture, pull and release (SCPR) mechanism, whereby Cdc12p nucleates actin filaments as Myo2p pulls actin filaments, thus producing the force required to pull the individual nodes into the ACR [28,34,35]. Such an assembly of the ACR from node precursors is schematically described in Figure 2.

### 2.3. Anchorage of the ACR to the Plasma Membrane

In the assembled ACR, the Myo2p tails and Cdc15p anchor to the plasma membrane, with the Myo2p heads, Myp2p and the bundle of actin filaments localizing 60 nm away from the plasma membrane [36]. It is suggested that this organization connects the bundle of actin filaments to the plasma membrane [37]. Cdc15p next recruits Cdc12p to the ACR, and this interaction is thought to be essential for ACR organization and stability [38]. 

The phospho-status of Cdc15p influences its ability to bind the plasma membrane, with the phosphorylation of Cdc15p by Pom1p inhibiting its binding to the plasma membrane at the cell tips [39]. Additionally, the p21-activated protein kinase (Pak1p), another polarity kinase, was found to regulate the function of Mid1p and Cdc15p [40]. Cdc15p has three regulatory components: an N-terminal Fre/Cip4 homology Bin/Amphiphysin/Rvs domain (F-BAR), a medial intrinsically disordered region (IDR) and a C-terminal Src homology 3 domain (SH3). While the F-BAR domain enables protein oligomerization and concentration on the plasma membrane to scaffold protein assemblies resulting in membrane deformation [41], it was recently found that the phosphorylation of Cdc15p induces the separation of the Cdc15p IDR region resulting in an inhibition of Cdc15p phase separation, and the formation of condensate on the plasma membrane [42]. 

Moshtohry et al. [43] recently used laser ablation, a technique based on photodamage in which cellular structures could be degraded using a focused pulsed laser, to investigate the mechanical role of Cdc15p during *S. pombe* cytokinesis and found that the ACR recoils after being severed. However, this recoil profile was greater and slower in the ablated ACR of Cdc15-depleted cells, suggesting that the loss of Cdc15p decreases the stiffness of the ACR material. Furthermore, another F-BAR protein, Imp2p, was found to contribute to the stiffness of the ACR [37].

## 3. ACR Constriction Is Coordinated with Septation in Fission Yeast

Unlike in mammalian cells, where cytokinesis only implies the formation of a medial ACR, cytokinesis in *S. pombe* additionally requires the formation of a cell-wall-like structure named the septum, physically separating the two daughter cells at the division site. After ACR assembly and constriction, the septation initiation network (SIN) mediates the synapses of amphid defective (SAD) kinase, and Cdr2p, dispersal from the cell cortex into the cytoplasm and septation is coordinated with the ACR constriction [44]. The septum is composed of three layers: a middle primary septum layer that is later digested by the end of cytokinesis, and two flanking secondary septa layers which remain intact to form the new cell walls of the separated daughter cells, reviewed by Pérez et al. [45] and Hercyk et al. [46].

*S. pombe* septation occurs in three stages. First, the deposition of the septum cell wall structure material is carried out through membrane trafficking events where secretory vesicles deliver the septum beta-glucan synthase 1, Bgs1p [47]. The localization of the cell-wall-building enzymes including Bgs1p depends on both Cdc42p and Cdc15p [48,49]. Second, during anaphase B, the ACR constricts at a slower rate as the septum ingression is initiated; however, after the delivery of the alpha-glucan synthase 1, Ags1p and Bgs4p, the rate of constriction and septum ingression is increased [50]. Third, after the ACR constricts, exocytosis leads to the delivery of glucanases and digestion of the primary septum, leaving the two daughter cells each with a new cell wall (secondary septa) [51].

The mechanical properties of *S. pombe* imply a high turgor pressure of the cytoplasmic fluid encased in a semi-permeable plasma membrane and a cell wall [52]. Proctor et al. [53] suggested that the ingression of the plasma membrane from the medial cortex requires large mechanical stress to counter the turgor pressure. 

Recently, Chew et al. [54] investigated the effects of the extracellular glycan matrix on ACR contraction. They used a fission yeast thermosensitive mutant *cps1-191*, defective in beta-glucan-synthase septum synthesis. This mutant was found to arrest with the contractile ring unable to constrict at the restrictive temperature. However, upon the weakening of the extracellular glucan matrix, ACR constriction and membrane ingression are enabled, indicating that the extracellular glycan matrix restricts membrane ingression and ACR constriction, and that this restriction is relieved upon the remodeling of the extracellular matrix and septum synthesis to facilitate ACR constriction.

Septation is coupled to ACR constriction, with the interaction between Bgs1p and the ACR is mediated by the paxillin Pxl1p [55]. Furthermore, Pxl1p accumulation during septum formation is thought to be mediated by an interaction with both the N-terminal F-BAR [38] and C-terminal SH3 domains of Cdc15p [39]. The mechanism of the engagement of Pxl1p to both the distal domains of Cdc15p is not well understood. However, a recent study demonstrated this interaction and found that Pxl1p binds to the Cdc15p F-BAR cytosolic domain [56].

During ACR constriction, a recently characterized Anillin homologue, Mid2p, is involved in the formation of a septin-based ring [57]. This ring is composed of septin complexes, which appear as a dense network of puncta attached to the plasma membrane [58], and other proteins including the digestive endo-(1,3)-beta-glucanase, Eng1p and the endo-(1,3)-alpha-glucanase, Agn1p [59].

Septins are not essential for cytokinesis in *S. pombe*. Nevertheless, they are critical components during cell separation in the later stages of cytokinesis, as the absence of the core septin component, Spn1p, leads to the advanced assembly of the ACR, a significant decrease in the ACR constriction and the premature formation of the septin ring [58]. Septins have an established role in recruiting glucanase enzymes to mediate cell separation; however, their function during the earlier stages of cytokinesis is unclear. A recent model suggests a tight connection between septins and the ACR whereby septins regulate the accumulation and maintenance of Sid2p, Bgs1p and Ags1p at the medial region [58]. 

During cell separation, turgor pressure pushes out the septum cell wall and at the new cell tips, generating a rounded shape through a mechanism that inflates the elastic cell wall in the absence of cell growth pressure [60]. When completed, primary septum digestion splits the two daughter cells apart. Septation coordination with ACR constriction is schematically described in Figure 3.

## 4. Insights into the Role of ESCRT Machinery during Cytokinesis

### 4.1. The Role of ESCRTs in Mammalian Cytokinesis

The endosomal sorting complex required for transport (ESCRT) genes were first characterized in *S. cerevisiae* as vacuolar protein sorting (*VPS*) genes, as class E *Vps* mutants were found to impair the sorting of vacuolar proteins [61,62,63]. An additional ESCRT component, the AAA-type ATPase VPS4, was revealed to regulate the membrane association of the VPS protein complex to control endosome function [64]. Subsequently, the ESCRT eukaryotic membrane remodeling machinery was shown to play major roles in other important cellular processes including the release of intralumenal vesicles (ILVs) during multivesicular body (MVB) formation [65], and the separation of daughter cells during cytokinesis [66]. In this section, we discuss the dynamics of the ESCRT/VPS4 membrane remodeling during cytokinesis as its role in other remodeling processes has been reviewed elsewhere [65,67,68]. 

The ESCRT machinery is composed of four complexes: ESCRT-0, ESCRT-I, ESCRT-II and ESCRT-III, with the latter complex required for all ESCRT activities performing the core function of membrane remodeling. ESCRT-III, being the most evolutionary ancient ESCRT member, has a unique ability to catalyze the fission of membrane necks from their luminal side. The ESCRT-III family is composed of 12 proteins named charged multivesicular body proteins (CHMPs) and plays critical roles during cytokinesis. 

Numerous studies have dissected the behavior of ESCRTs during cytokinesis, reviewed by Nähse et al. [69] and McCullough et al. [70]. Briefly, during mammalian cell cytokinesis, the ingression of the cleavage furrow creates a thin intracellular bridge connecting the two daughter cells named the midbody. ESCRT-I protein tumor-susceptibility gene 101 (TSG101) and ESCRT-I adaptor apoptosis-linked gene 2-interacting protein X (ALIX) interact with the centrosomal protein of 55 kDa (CEP55) at the midbody [66]. The ESCRT-III protein CHMP4B is then recruited to the site of constriction on both sites of the midbody center, the polymerization of ESCRT-III into a spiral on the side of the microtubule bridge occurs and VPS4 next catalyzes the rearrangement of the spiral [71,72]. 

ESCRT-III proteins mediate filament formation via their helical core domains, in addition to their C-terminal tail domain including the short peptide elements named MIT-interacting motifs (MIMs). MIM sequences bind cofactors containing microtubule-interacting and trafficking (MIT) domains; for example VPS4 contains an archetypal MIT domain which binds promiscuously to ESCRT-III filaments, promoting the assembly of VPS4 hexamers and ATPase activity [73]. VPS4 is the only nucleotide hydrolase in the ESCRT machinery [64], and it functions by pulling ESCRT-III subunits out of the filament to unfold their polypeptide chain through a central pore; this is achieved via cycles of binding ATP, hydrolysis and ADP release [74]. In earlier studies, VPS4 was detected at the site of division only during the last stages of abscission [71]. However, more recently, VPS4 was also detected during the early stages of abscission and found to be required for ESCRT-III turnover at the midbody [75]. The role of the ESCRTs during mammalian cell cytokinesis is schematically described in Figure 4.

### 4.2. The Role of ESCRTs in Fission Yeast Cytokinesis

It has been established that ESCRT-III recruits VPS4 to remodel filaments driving the midbody constriction in mammalian cells [72,75,76,77]. However, the exact mechanism of how ESCRT-III and VPS4 catalyze membrane fission during abscission in other model organisms including *S. pombe* remains unclear.

Sequencing the *S. pombe* genome revealed that the ESCRT proteins are conserved in this yeast species [78,79], and it was later confirmed that Class E *Vps* proteins regulate vesicle-mediated protein sorting [80]. ESCRTs are also involved in spindle pole body (SPB) dynamics, however, and there are many other unexplored functions of ESCRTs in *S. pombe*, including their role in sealing newly formed nuclear envelopes and during cytokinesis. ESCRTs demonstrate strong genetic interactions with SPB proteins and transmembrane nucleoporins, suggesting a role of ESCRTs in nuclear remodeling [81]. An interesting study explored the ESCRT-mediated nuclear envelope closure in mammalian and *S. pombe* cells and showed that ESCRT-III mutants and *vps4*Δ cells display defective SPB amplification and severe defects in nuclear integrity and morphology [82]. 

Emerging evidence suggests an important role of the ESCRT machinery during *S. pombe* cytokinesis. Septation is defective in individual *S. pombe* ESCRT mutants and *vps4*Δ cells remain attached during septation and fail to separate [83]. This phenotype implies an important role for Vps4p during abscission. However, it is important to determine Vp4p interacting partners to explore this function. A genetic screen investigated genetic interactions between each of the *S. pombe* ESCRT genes and the anillin-encoding *mid1* gene. Amongst all the ESCRTs, only the *vps4* gene showed genetic interactions as a *mid1*Δ *vps4*Δ double mutant demonstrated synthetic lethality [22]. Furthermore, the Vps4p protein has been suggested to interact directly or indirectly with the C-terminal domain of anillin/Mid1p, probably through the lipid-binding PH domain. But how is this coordination carried out if Mid1p leaves the ACR before the onset of abscission when Vps4p is activated? 

Until recently, it was not understood how Mid1p leaves the site of division upon the ACR constriction. However, part of this was explained when it was recently found that the phosphorylation of Mid1p by Sid2p leads to Mid1p’s cortical dissociation from the plasma-membrane-attached ACR [84]. It is possible that a direct or indirect coordination between the ATPase Vps4p and the SIN kinase Sid2p might regulate Mid1p’s disassociation, leading to its nuclear re-localization during the later stages of cytokinesis. Investigating the co-localization of Mdi1p, Vps4p and Sid2p in *S. pombe* could be informative in this regard. Additionally, it is known from time lapse imaging experiments that *S. pombe* cells lacking Sid2p phospho-sites cause Mid1p to maintain cortical attachment even at the onset of ACR constriction [84]. Therefore, similarly, tracking the localization of Mid1p in *vps4*Δ mutant cells throughout the cell cycle could lead to a clearer understanding of the ESCRT machinery dynamics in regulating *S. pombe* cytokinesis.

## 5. Concluding Remarks

Studying the function of gene products and their localization through the cell cycle in *S. pombe* has been a powerful way to understand eukaryotic cell division. We now know that division in *S. pombe* is excluded from the cell tips through a negative Pom1p signal, and the future site of division is determined by the centrally positioned nucleus through positive Mid1p signals which form medial cortical nodes. These nodes mature into the ACR and, as the ACR constricts, the septum is synthesized to form the future cell wall structure of the daughter cells. The digestion of the medial septal material executes cell division leading to two daughter cells. 

However, this vibrant research field harbors important unsolved questions, especially with advances in live cell imaging, fluorescent macromolecules, and the integration of different research fields including cell mechanics and mathematical modelling.

Anillin-like proteins have established roles during cytokinesis in eukaryotes. The mechanisms of cell division site localization in both human anillin and *S. pombe* Mid1p depend on their phospho-regulation. It would be interesting to better understand how phosphorylation affects protein function with regard to specific aspects of cytokinesis including scaffold proteins with plasma-membrane-anchoring properties, and the stable attachment of the ACR to the plasma membrane. The phospho-regulation of the two ACR scaffold proteins, Mid1p and Cdr2p, revealed a regulatory mechanism through their interaction with the plasma membrane. Additional experiments with plasma membrane lipids and advances in proteomics research could be informative.

Fluorescence microscopy demonstrated that ESCRTs and VPS4 co-localize at the abscission site during mammalian cytokinesis, and their role to regulate this process was extensively studied. We know that the ESCRT machinery is also required for cytokinesis in *S. pombe*, but what are the specific protein interactions controlling this process? Vps4p is a strong candidate, and we have initial evidence of coordination between Vps4p and Mid1p. The discovery of a Mid1p network of interactions with different classes of proteins including the polo-like kinase, SIN kinase and PAK kinase implies the importance of Mid1’s regulation and suggests its potential regulation by the ESCRT machinery.

This review discusses *S. pombe* cytokinesis and presents a visualized picture of the current understanding of the proteins and processes that control this important event. It is anticipated that these mechanisms will be conserved across all eukaryotes and so will be informative about human cells and disease conditions, where they are defective. 

## Figures and Tables

**Figure 1 jof-10-00154-f001:**
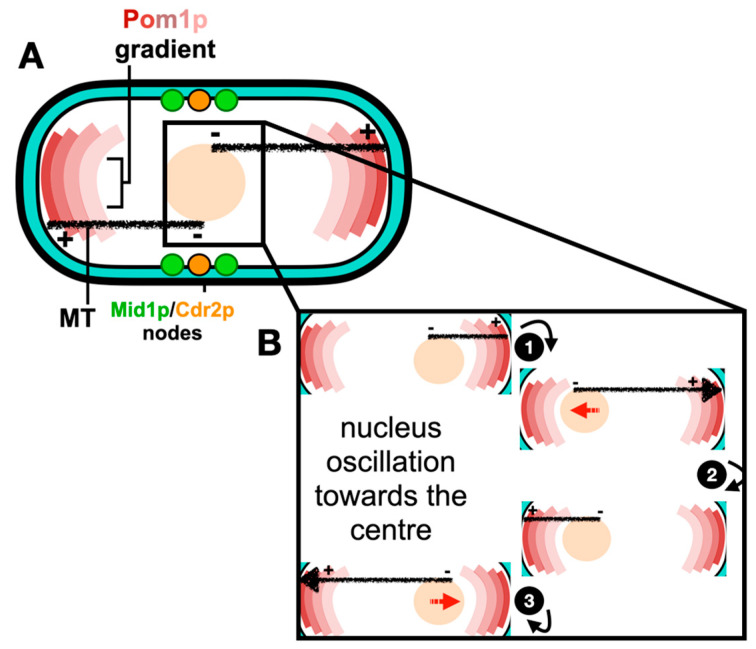
Medial positioning of the ACR in fission yeast. (**A**) Pom1p gradient at the cell tips restricts the division site to the cell center. Mitotic entry is controlled by Pom1p phosphorylation of Cdr2p, preventing Cdr2p from binding the plasma membrane and the subsequent formation of cortical nodes. Upon mitotic entry, both active Mid1p and Cdr2p scaffold the formation of medial cortical nodes. (**B**) Upon microtubule–cortex contact, polymerization at the microtubule plus end generates a pushing force (large arrowhead) (1) that displaces the nucleus in the opposite direction (nuclear movement demonstrated by red arrows) (2). The antiparallel direction of the microtubule bundle ensures that, over time, the nucleus oscillates back and forth toward the center (3). References within the main text.

**Figure 2 jof-10-00154-f002:**
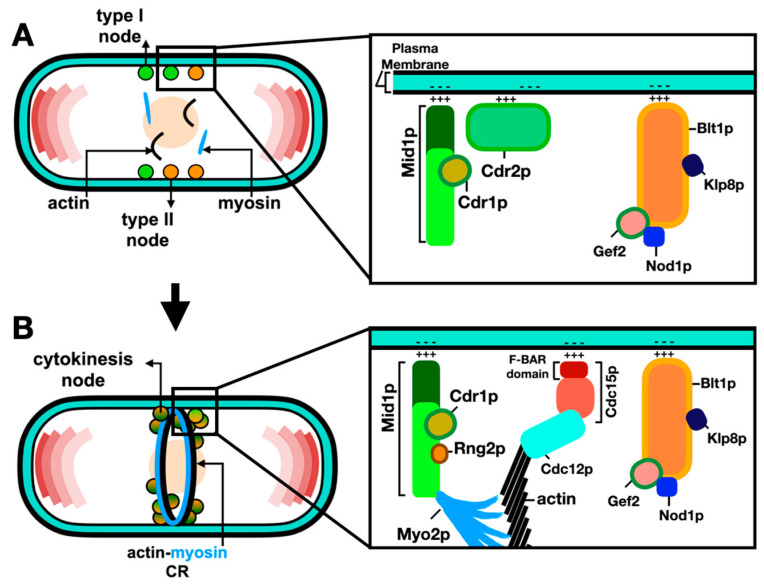
Assembly of the ACR from node precursors. (**A**) During interphase, type 1 “stationary” (green: Mid1p, Cdr1p and Cdr2p) nodes and type 2 “anchoring” (orange: Blt1p, Klp8p, Gef2p and Nod1p) nodes bind the plasma membrane and scaffold other cytokinesis proteins. (**B**) Coalescence of type 1 and type 2 nodes leads to their maturation into cytokinesis nodes (green–orange gradient). Maturation of cytokinesis nodes leads to the recruitment of Myo2p, Cdc15p and Cdc12p and nucleation of actin filaments. Interactions between myosin-II and actin promote ACR formation. References within the main text.

**Figure 3 jof-10-00154-f003:**
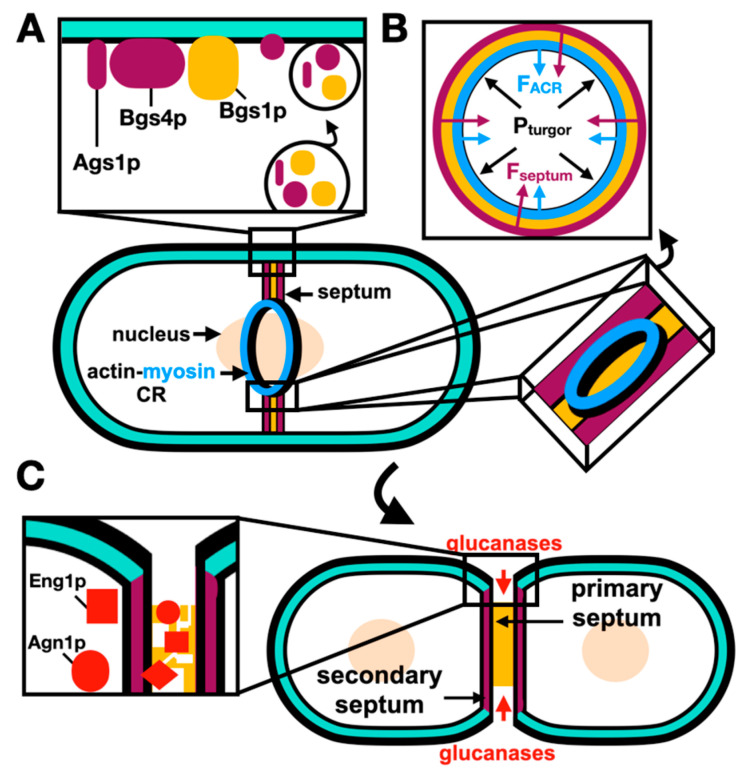
Septum biosynthesis is coupled to contractile ring constriction. (**A**) A primary septum (yellow) flanked by two secondary septa layers (purple) forms behind the ACR as it constricts. Deposition of the septum material by membrane trafficking through secretory vesicles carrying the septum glucan synthases: secondary septum proteins Ags1p, Bgs4p and primary septum protein Bgs1p. (**B**) Cross-section of the deposited septum and constricting ring with internal turgor pressure (black), opposed by the inward forces of the ACR constriction (blue) and septation (purple). (**C**) Following the ACR closure, the primary septum layer is degraded by glucanases (red: Eng1p and Agn1p), which are delivered to the septum via a septin-based pathway, the secondary septa represent the cell wall material at the new cell tips. Derived from Proctor et al. [53]; references within the main text.

**Figure 4 jof-10-00154-f004:**
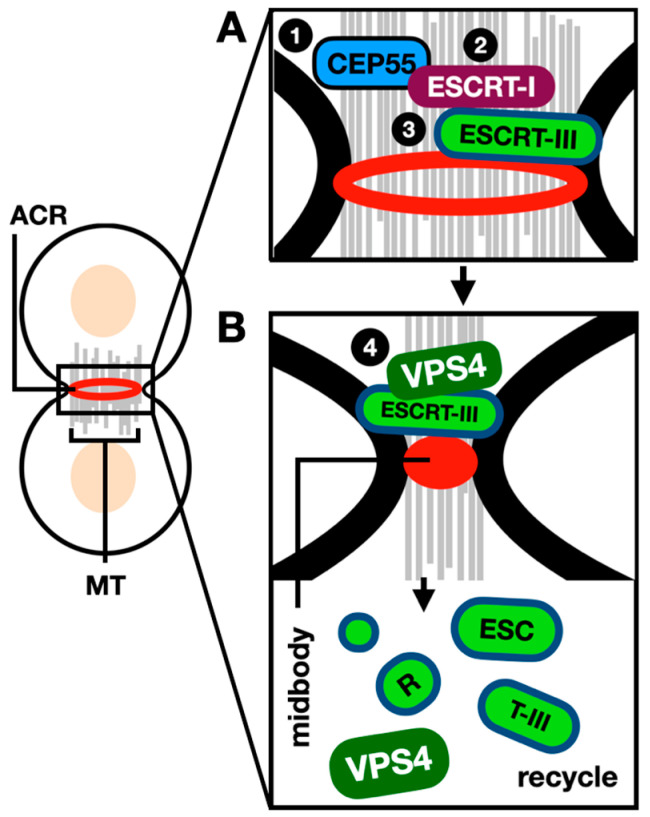
The role of ESCRTs during mammalian cytokinesis. During mammalian cytokinesis, a cleavage furrow forms medially between the two dividing cells, while the ACR constricts to create the midbody. (**A**) Sequential recruitment (steps 1–3) of the ESCRT-I subunits/factors leads to the recruitment of ESCRT-III polymers forming a spiral at the constricted midbody. (**B**) The ATPase VPS4 is then recruited to the midbody (step 4) where it disassembles the ESCRT-III polymers, this ensures ESCRT-III subunits are recycled at the division site with such fission reaction leading to the separation of daughter cells. ACR: actin–myosin contractile ring. MT: microtubules. References within the main text.

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
