# Peer review of "Processes Controlling the Contractile Ring during Cytokinesis in Fission Yeast, Including the Role of ESCRT Proteins"

_jof, 2024, doi:10.3390/jof10020154_

Round 1

Reviewer 1 Report

Comments and Suggestions for Authors

The manuscript by Rezig IM et al. is a review that summarizes the current state of our understanding of the mechanisms that control the dynamics of the contractile ring during cytokinesis in S. pombe. The opening paragraph describes fission yeast as a valuable model to study mechanisms of cytokinesis that are conserved in higher eukaryotes and therefore relevant to human cells. The authors divided the remaining text into sections that follow the main events during cytokinesis starting with the mechanisms that establish the positioning of the cytokinetic machinery and ending with the final cell separation and the role of the ESCRT complex in this process. The manuscript concludes with a brief Concluding Remarks section. Overall, this is a well-crafted review supported with informative schematics that depict various mechanisms related to consecutive stages of cytokinesis. Some editing would be helpful especially in some parts of the manuscript. Below are my specific comments and suggestions for changes that may help to further improve this manuscript. 

1.     The text would benefit from some editorial work to improve the grammar. I included some specific suggestions below, but more thorough editing would be beneficial.

2.     The first sentence of the Abstract needs to be rewritten and improved.

3.     Ln 13: equally sized

4.     Ln 14 - delete "such"

5.     Ln 15 and the remaining text: “actomyosin ring”

6.     Ln 19: "the abscission"

7.     Ln 20-21: "the two phases of cytokinesis, ACR assembly and constriction, and their coordination with septation"

8.     Ln 22: replace "with" with "and emphasize..."

9.     Ln 35: replace "stems" with " has benefited significantly from studies..."

10.  Ln 53: delete "cytokinetic"

11.  Ln 56: replace "during" with "following"

12.  Ln 64: replace "and is applied to" with "acts on"

13.  Ln 68: I suggest deleting Alternatively and starting the sentence with "The "end or tip" location is ..."

14.  Ln 78: released from the nucleus? not resealed.

15.  Ln 203: ...regulates...

16.  Ln 230: performing

17.  Ln 308-313: Here the Authors summarize the basic concept of cytokinesis, common to most yeast cells. I am not sure this should be the main conclusion as this has been known for decades. Instead, perhaps the Authors could summarize all the most recent advancements and/or, like my other comment states, the Authors could point out to questions that remain still unanswered.

18.  The Concluding remarks section would benefit from a summary of what are the major gaps in knowledge still remaining and what are specific questions that need to be addressed. The only specific aspect the Authors mention here are the ESCRT and VPS4. One aspect that could be mentioned is the role of septins in cytokinesis, a subject that remains poorly understood but also which has been de-emphasized in this review.

19.  Ln 318-319: the last sentence should be rewritten.

Comments on the Quality of English Language

Some editing would be needed

Author Response

Reviewer #1

  1. The first sentence of the Abstract needs to be rewritten and improved.

We have re-structured the first sentence of the Abstract, as asked.

  1. Ln 13: equally sized

This was corrected in text.

  1. Ln 14 - delete "such"

This was corrected in text.

  1. Ln 15 and the remaining text: “actomyosin ring”

This was corrected in text.

  1. Ln 19: "the abscission"

This was corrected in text.

  1. Ln 20-21: "the two phases of cytokinesis, ACR assembly and constriction, and their coordination with septation"

This was corrected in text.

  1. Ln 22: replace "with" with "and emphasize..."

This was corrected in text.

  1. Ln 35: replace "stems" with " has benefited significantly from studies..."

This was corrected in text.

  1. Ln 53: delete "cytokinetic"

This was corrected in text.

  1. Ln 56: replace "during" with "following"

This was corrected in text.

  1. Ln 64: replace "and is applied to" with "acts on"

This was corrected in text.

  1. Ln 68: I suggest deleting Alternatively and starting the sentence with "The "end or tip"

This was corrected in text.

  1. Ln 78: released from the nucleus? not resealed.

This was corrected in text.

  1. Ln 203: ...regulates...

This was corrected in text.

  1. Ln 230: performing

This was corrected in text.

  1. Ln 308-313: Here the Authors summarize the basic concept of cytokinesis, common to most yeast cells. I am not sure this should be the main conclusion as this has been known for decades. Instead, perhaps the Authors could summarize all the most recent advancements and/or, like my other comment states, the Authors could point out to questions that remain still unanswered.

We agree, and we have re-structured the concluding remarks, as asked.

  1. The Concluding remarks section would benefit from a summary of what are the major gaps in knowledge still remaining and what are specific questions that need to be addressed. The only specific aspect the Authors mention here are the ESCRT and VPS4. One aspect that could be mentioned is the role of septins in cytokinesis, a subject that remains poorly understood but also which has been de-emphasized in this review.

We have noticed that this comment has also been asked by reviewer #3, and so we have emphasized the role of septins during cytokinesis in the text.

  1. Ln 318-319: the last sentence should be rewritten.

We agree, and we have re-structured the concluding remarks, as asked.

Reviewer 2 Report

Comments and Suggestions for Authors

Fission yeast has been utilized to elucidate the mechanisms of eukaryotic cytokinesis. This review article summarizes the contribution of this model organism in this field, and provides the future prospect for further research into cytokinesis. Overall, this article concisely describes the current understanding of the mechanisms regulating cytokinesis, and successfully introduces fission yeast as a powerful tool to study cytokinesis. Nevertheless, I recommend some revisions before publication, since I feel that some parts of this manuscript are difficult to understand and lack enough explanations as described below.

p3, line 90–92

It would be more interesting to describe the detailed mechanism of the mitotic entry regulated by Pom1p. As a cell grows, the concentration of Pom1p should be reduced around the equatorial region. Is this reduction involved in the regulation of mitotic entry? 

p3, line 99–103

The description regarding the two types of nodes is not consistent with Figure 2 in some respects. The text describes that Rng2p is a component of anchoring nodes, and Mid1p and Cdr1p are the components of stationary nodes, but they are in the same complex in Figure 2B. In addition, whereas the text describes that anchoring nodes anchor the ends of myosin-II, stationary nodes are connected to myosin-II in Figure 2B. Furthermore, anchoring nodes containing Blt1p in Figure 2A are totally different from anchoring nodes containing Cdc15p and Cdc12p in Figure 2B. Do anchoring nodes change radically during the cytokinesis process or is only a part of anchoring nodes drawn in each figure?

p5, line 154–155

Cdc15p is not essential for the ACR formation, isn’t it? Are there other proteins having the equivalent function to that of Cdc15p?

p5, line 191–194

I do not understand the logic of this part. Does decreased turgor pressure inhibit the ACR constriction in the mutant? Why does that suggest the involvement of extracellular component in the ACR constriction? More explanation is needed.

p5, line 195–197

How does turgor pressure drive abscission? What is the relationship between turgor pressure and hydrolysis of the primary septum described in the following sentences.

Figure 3

In the magnified illustration around the cell wall in Figure 3A, the proteins indicated by purple and yellow, respectively, are both Bgs4p. In addition, the magnified illustration of ACR and the septum is difficult to understand. It looks that acto-myosin forms a ring on the three-layered septum at this region. Is it correct?

p7, line 242–245

Describe clearly the function of ESCRT-III for abscission. Does the spiral chain squeeze out the connecting bridge or is the regulation of vesicle trafficking involved in abscission?

Minor comments

p2, line 78, resealed -> released

p5, line 163, The explanation of Cdr2p should be added when it first come out.

p5, line 186, synthesis in important -> synthesis is important

Author Response

Reply to Reviewer 2

p3, line 90–92: It would be more interesting to describe the detailed mechanism of the mitotic entry regulated by Pom1p. As a cell grows, the concentration of Pom1p should be reduced around the equatorial region. Is this reduction involved in mitotic entry? 

We have re-structured this section to include more details about mitotic entry regulation by Pom1p and its gradient at cell tips.

p3, line 99–103: The description regarding the two types of nodes is not consistent with Figure 2 in some respects. The text describes that Rng2p is a component of anchoring nodes, and Mid1p and Cdr1p are the components of stationary nodes, but they are in the same complex in Figure 2B. In addition, whereas the text describes that anchoring nodes anchor the ends of myosin-II, stationary nodes are connected to myosin-II in Figure 2B. Furthermore, anchoring nodes containing Blt1p in Figure 2A are different from anchoring nodes containing Cdc15p and Cdc12p in Figure 2B. Do anchoring nodes change radically during the cytokinesis process or is only a part of anchoring nodes drawn in each figure?

We appreciate this important comment. The description of the two types of nodes in the text is demonstrated with more clarity, we also edited Figure 2 and the legend accordingly.

p5, line 154–155: Cdc15p is not essential for the ACR formation, isn’t it? Are there other proteins having the equivalent function to that of Cdc15p?

We have re-structured the first sentence of the Abstract.

p5, line 191–194: I do not understand the logic of this part. Does decreased turgor pressure inhibit the ACR constriction in the mutant? suggest the involvement of extracellular component in the ACR constriction? More explanation is needed.

We appreciate this important comment. The logic of this part was demonstrated with more clarity in the text.

p5, line 195–197: How does turgor pressure drive abscission? What is the relationship between turgor pressure and hydrolysis of the primary septum described in the following sentences.

We appreciate this important comment. The logic of this part was demonstrated with more clarity in the text.

Figure 3: In the magnified illustration around the cell wall in Figure 3A, the proteins indicated by purple and yellow, respectively, are both Bgs4p. In addition, the magnified illustration of ACR and the septum is difficult to understand. It looks that acto-myosin forms a ring on the three-layered septum at this region. Is it correct?

We appreciate this important comment. Septum biosynthesis is coupled to the contractile ring formation, a primary septum flanked by two secondary septa layers forms behind the ACR as it constricts. We edited Figure 3 and further clarified the legend accordingly.

p7, line 242–245: Describe the function of ESCRT-III. Does the spiral chain squeeze out the connecting bridge or is the regulation of vesicle trafficking involved in abscission?

We appreciate this important comment. The mechanism of ESCRT-III function is further clarified in the text.

p2, line 78, resealed -> released

This was corrected in text.

p5, line 163, The explanation of Cdr2p should be added when it first come out.

This was corrected in text.

p5, line 186, synthesis in important -> synthesis is important

This was corrected in text.

Reviewer 3 Report

Comments and Suggestions for Authors

In this short review Regiz and collaborators describe some aspects of S. pombe cytokinesis.

The paper is well written and deserves publication after minor corrections.

Specific points:

-              It is not very clear the correlation between Pom1 gradient and its role in phosphorylating its substrates (lines 69-73). Does Pom1 phosphorylate Cdr2 in the middle of the cell where its abundance is low? Why? And what about Cdc15 phosphorylation by Pom1? (lines138-140). The authors should explain better and add a scheme in which phosphorylation events are described.

-              Figure 2: some proteins are not described in the legend.

-              Septins are crucial proteins in cytokinesis and they are only nominated at line 200, the authors should describe better their role in cytokinesis.

-              Figure 3: some proteins are not described in the legend.

Comments on the Quality of English Language

a

Author Response

It is not very clear the correlation between Pom1 gradient and its role in phosphorylating its substrates (lines 69-73). Does Pom1 phosphorylate Cdr2 in the middle of the cell where its abundance is low? Why? And what about Cdc15 phosphorylation by Pom1? (lines138-140). The authors should explain better and add a scheme in which phosphorylation events are described.

We have re-structured this section to include more details about Cdr2 phosphorylation by Pom1p.

Figure 2: some proteins are not described in the legend.

This was corrected in text.

Septins are crucial proteins in cytokinesis, and they are only nominated at line 200, the authors should describe better their role in cytokinesis.

We have noticed that this comment has also been asked by reviewer #1, and so we have emphasized the role of septins during cytokinesis in the text.

Figure 3: some proteins are not described in the legend.

This was corrected in text.